# Validation of Recycled Nanofiltration and Anion-Exchange Membranes for the Treatment of Urban Wastewater for Crop Irrigation

**DOI:** 10.3390/membranes12080746

**Published:** 2022-07-29

**Authors:** Anamary Pompa-Pernía, Serena Molina, Amaia Lejarazu-Larrañaga, Junkal Landaburu-Aguirre, Eloy García-Calvo

**Affiliations:** 1IMDEA Water Institute, Avenida Punto Com, 2, 28805 Alcalá de Henares, Madrid, Spain; serena.molina@imdea.org (S.M.); amaia.ortiz@imdea.org (A.L.-L.); junkal.landaburu@imdea.org (J.L.-A.); eloy.garcia@imdea.org (E.G.-C.); 2Chemical Engineering Department, University of Alcalá, Ctra. Madrid-Barcelona Km 33.600, 28871 Alcalá de Henares, Madrid, Spain

**Keywords:** circular economy, membrane recycling, nanofiltration membranes, nanofiltration, anion-exchange membranes, electrodialysis, wastewater treatment

## Abstract

One of the alternative sources to tackle the problem of water shortage is the use of reclaimed water from wastewater treatment plants for irrigation purposes. However, when the wastewater has a high conductivity value, it becomes unusable for crop irrigation and needs a more specific treatment. In this work, recycled nanofiltration (rNF) membranes and anion-exchange membranes (rAEMs) obtained from end-of-life RO membranes were validated to evaluate their application capability in saline wastewater treatment. The use of recycled membranes may represent an advantage due to their lower cost and reduced environmental impact associated with their production, which integrates membrane-based technology into a circular economy model. Both recycled membranes were tested in crossflow filtration and electrodialysis (ED) systems. The results of the rNF membrane showed a high selective rejection of divalent ions (SO_4_^2−^ (>96%) and Ca^2+^ and Mg^2+^ (>93%)). In the case of the ED process, the comparison between rAEMs and commercial membranes showed an appropriate demineralization rate without compromising the power consumption. Finally, the quality of both system effluents was suitable for irrigation, which was compared to the WHO guideline and validated by the 7-week lettuce crop study.

## 1. Introduction

Water scarcity is a big problem because the demand for the available water exceeds the conventional water resources due to population growth, climate changes, and ongoing industrialization. Under this context, the use of reclaimed water from wastewater treatment plants (WWTPs) as a source of irrigation water is an important strategy to ensure water security in many regions experiencing water scarcity issues [1]. The advantage of water reuse is directly connected to the avoidance of using drinking water for irrigation, reducing the over-extraction of surface and groundwater, and decreasing its dependence on climate change by using sustainable water resources [2]. However, it is estimated that around 5% of the total worldwide influent of WWTPs comprises saline and hypersaline wastewater quality [3]. In agriculture and water studies, electrical conductivity (EC) is used as an indicator of soil and water salinity. The EC values above 3000 µScm^−1^ are considered high values in water for irrigation purposes, representing a high risk for the yield of crops and for the structure and permeability of the soil [4]. Very high values of EC (hardness of water) in WWTPS effluents are commonly provided by various sources such as seawater intrusion, aquaculture, agriculture, and saline wastewater dischargement of various industries (i.e., petroleum and gas extraction, leather manufacturing) [5]. If water with high conductivity is applied to soil, it could have negative consequences such as reduced plant productivity, crop failure, and in extreme cases, death of vegetation [6,7]. Therefore, reclaimed water for agriculture, particularly for irrigation, should obey water quality guidelines and requires the development of a proper water management strategy. Consequently, to improve the quality of the WWTP secondary-treated wastewater effluents, advanced technologies must be applied.

Among the different technologies, membrane-based technologies are especially attractive for saline wastewater regeneration due to their intrinsic desalination capability, flexibility, and high permeate quality. Nor Naimah et al. [5] summarized the different applications of membrane-based processes in saline wastewater treatment. Within membrane processes operated under pressure-driven mode, there is microfiltration (MF), ultrafiltration (UF), nanofiltration (NF), and reverse osmosis (RO). MF and UF membranes are normally used to remove high-molecular-weight compounds (e.g., particulate matter and bacteria) from the water as the sole treatment process or pre-treatment for other processes due to the larger pore size [8,9], which implies that they fail to remove dissolved ions. On the contrary, NF and RO membranes with a tighter pore size are compelling for the separation of inorganic salts and small organic molecules. RO has been known for its capability to reject almost all impurities in the water, which leads to the use of RO to desalinate seawater and brackish water to produce drinking water. Compared to RO, the property of the charged surface of NF allows reducing hazardous electroconductivity values in wastewater treatment by exerting electrostatic repulsion toward multivalent ions [10]. Those characteristics of NF membranes have led to the implementation of NF as a tertiary treatment for olive oil mill wastewater, for removing dyes in the textile industry, and as a treatment to recover useful resources from the wastewater (e.g., fractional process of humic substances, sulphates from tannery wastewater) [5]. Nor Naimah et al. [5] also highlighted electrodialysis (ED) as another promising membrane-based separation that can be used in saline wastewater treatment. The ED technology is essentially different compared with pressure-driven processes since it is an electrically driven membrane process, where cation- and anion-exchange membranes are alternately placed in a stack, achieving the separation of cations and anions, respectively [11]. However, its application in wastewater treatment such as oil and gas effluent, leachate, textile, and tannery wastewater has also been reported.

Despite the promising performance of membrane technologies, they still deal with drawbacks such as fouling issues and limited lifespan, which increase not only the operational cost of the technologies but also the generation of waste sources. In this regard, the use of recycled membranes may represent an advantage due to their lower cost and the reduced environmental impact associated with their production. In this sense, membrane technology could be integrated into a circular economy model, in concordance with the European Green Deal [12] and the priority order of waste management (prevention, reuse, recycling, recovery, and disposal) regulated in the European Directive 2008/98/EC.

An innovative approach in membrane recycling was proposed since the introduction of the concept of transformation of end-of-life membranes into ultrafiltration membrane in 2002 [13]. Different studies have given a second life to discarded RO membranes by the transformation of spiral wound module configuration into recycled NF- and UF-like membranes [14,15,16]. Those recycled UF membranes were tested to treat wastewater [14] and gray water reclamation [17], while recycled NF membranes were tested for brackish water treatment [18]. In addition, other studies proposed the deconstruction of the spiral wound configuration to enable the individual management and valorization of the membranes and other plastic components of the RO module (e.g., polypropylene feed spacers) [19]. Under this last approach, anion-exchange membranes (AEMs) have been prepared by upcycling discarded RO membranes and have been validated as proof of concept in the ED process [20,21]. In that sense, the technical viability of the recycled AEM was shown to be viable in brackish water desalination experiments with a synthetic solution of NaCl, obtaining 84.5% of salt removal [20].

Even though there is a recognized potentiality of the recycled membranes, it is still required to investigate whether the recycled membranes have overall performance characteristics and application capabilities comparable with the performance of commercial membranes. In addition, wastewater regeneration is perfectly integrated within the circular economy definition, and hence, it is in very good alignment with the membrane recycling approach. Therefore, the present work aimed to make one further step in evaluating the implementation of recycled membranes previously developed by our research group (i.e., recycled NF membrane and recycled AEM) in NF and ED processes for saline urban wastewater (UWW) remediation to obtain water for crop irrigation. The quality of both system effluents was evaluated by comparing the quality of the water reused for irrigation, which is based on the guideline of the World Health Organization (WHO) [2]. To further validate the quality of the effluents, the obtained regenerated waters using recycled membranes were studied for the first time for lettuce cultivation.

## 2. Materials and Methods

### 2.1. Wastewater Sample

The feed used in this work was synthetic saline wastewater. The recipe was prepared based on the data analysis from an urban WWTP on the Levante coast in Spain, which confronted a salinity intrusion issue. The real UWW samples were collected every month over three years from the secondary clarifier tank of the plant, where wastewater is biologically treated by the conventional activated sludge. The UWW in this study contained Cl^−^ (1220 mg L^−1^), NO_3_^−^ (57.22 mg L^−1^), SO_4_^2−^ (263 mg L^−1^), Na^+^ (694 mg L^−1^), K^+^ (40 mg L^−1^), Ca^2+^ (204 mg L^−1^), and Mg^2+^ (99.9 mg L^−1^) with a pH of 7.16–7.85 and electrical conductivity (EC) of 4830–5237 µS cm^−1^.

### 2.2. Chemical Reagents

Sodium hypochlorite (NaClO, 14%), sodium hydrogen phosphate (NaH_2_PO_4_), sodium chloride (NaCl), calcium chloride dihydrate (CaCl_2_∙2H_2_O), magnesium sulfate heptahydrate (MgSO_4_∙7H_2_O), sodium hydrogen carbonate (NaHCO_3_), potassium nitrate (KNO_3_), sodium nitrate (NaNO_3_), calcium carbonate (CaCO_3_), magnesium chloride hexahydrate (MgCl_2_∙6H_2_O), potassium sulfate (K_2_SO_4_), sodium sulfate (Na_2_SO_4_), calcium sulfate dihydrate (CaSO_4_∙2H_2_O), boric acid (H_3_BO_3_, 0.5%), hydrogen peroxide (H_2_O_2_, 30%), and chloride acid (HCl, 0.1 M) were purchased from Scharlab S.L., Barcelona, Spain. The ultrapure water (Milli-Q) used in the experiments was obtained from Millipore, Molsheim, France, equipment (conductivity less than 0.055 µS cm^−1^).

Polyvinyl chloride (PVC, Mw 112,000 g mol^−1^) was supplied by ATOCHEM, Madrid, Spain. Tetrahydrofuran (THF) was purchased from Scharlab S.L. Amberlite^®^ IRA-402 (Cl^−^ form, total exchange capacity ≥ 1.0 mol L^−1^) was supplied by Merck Life Science, Darmstadt, Germany, S.L.U.

### 2.3. Preparation of Membranes

Following the protocols previously reported in [19,21], recycled nanofiltration membranes (rNF) and recycled anion-exchange membranes (rAEMs) were prepared [15,20]. For this purpose, the membrane coupons (~315 cm^2^) were extracted from discarded 8″ diameter RO membrane spiral wound module (TM720-400, Toray Industries, Inc., Osaka, Japan) by membrane autopsy explained elsewhere [15]. Then, the passive transformation protocol was followed [15] at different exposure doses (detailed below) to obtain rNF and recycled ultrafiltration (rUF) membranes (see Figure 1).

The oxidizing agent for membrane transformation was NaClO (14%), and the free chlorine concentration was analyzed using a Pharo 100 Spectroquant spectrophotometer before membrane exposure. The feed and permeate spacers from the end-of-life RO module were also reused for the NF and ED processes, as is specified in Section 2.4.1 and Section 2.4.2.

#### 2.3.1. Recycled Nanofiltration Membranes

The rNF membrane transformation was conducted using an exposure dose of 8000 ppm h of NaClO solution at room temperature (~21 °C). This exposure dose ensured the total elimination of the fouling and the partial elimination of the polyamide thin film layer of the discarded RO (Figure 1), as it has been widely investigated previously by our research group [15]. To ensure partial removal of the polyamide layer (PA) and check the achievement of rNF membrane properties, attenuate total reflectance Fourier transform infrared (ATR-FTIR) spectroscopy was used (Perkin-Elmer RX1 spectrometer) (Perkin-Elmer, Waltham, MA, USA) (Appendix A), and some characterizations tests were carried out (see the Appendix A).

#### 2.3.2. Recycled Anion-Exchange Membranes

The preparation of the rAEM was conducted following the procedure previously reported by Lejarazu-Larrañaga et al. [20]. Briefly, the first step was to obtain the rUF membranes for their use as mechanical support. The transformation into rUF membranes was conducted with an exposure dose of 500,000 ppm h NaClO solution at room temperature (~21 °C). It was verified that the thin film PA layer was completely removed by employing the ATR-FTIR technique (Appendix A). The polymeric mixture employed was prepared using the following chemicals: (i) PVC as a polymer blinder, (ii) THF as the solvent, and (iii) Amberlite^®^ IRA-402 as anion-exchange resin. Then, the membranes were prepared using a casting knife and extending an 800 µm thick polymeric mixture on the surface of the rUF membrane. Subsequently, the solvent was evaporated for 60 min at room temperature, and the membranes were finally immersed in a water bath at 20 °C. The performance of the rAEM was compared with a commercial AEM, in this case, Ralex^®^ AMH-PES from Mega a.s., Straz Pod Ralskem, Czech Republic. This membrane (Ralex^®^ AMH-PES) was selected as a reference as long as it had a heterogeneous structure, like the prepared rAEM, which included conducting and non-conducting regions [22]. Indeed, the type of the structure (heterogenous membrane) of the prepared membrane was observed by SEM and compared with a heterogenous Ralex^®^ AHM-PES membrane, which was reported previously and can be found in [20].

### 2.4. Membrane Performance in UWW Treatment

#### 2.4.1. Nanofiltration Experiments

Crossflow flat-sheet membrane system (from IBERLACT S.L., Madrid, Spain) used to perform the NF tests for the filtration of the synthetic UWW is represented in Figure 2a. The system has a high-pressure pump, a 25 L feed reservoir, and a tubular heat exchanger with a temperature controller. The NF membrane with an effective membrane filtration surface of 84 cm^2^ was placed into a flat-sheet stainless steel RO test cell, arranging the permeate and feed spacers from the discarded RO module in the same position as in the original module. The UWW feed and permeate conductivity were measured every 10 min as described in Section 2.6.

For the NF assays (based on recycled membranes), the coupon was compacted with a total UWW feed volume of 5 L, a flow rate of 3.9 L min^−1^, a temperature of 25 °C, and 10 bar transmembrane pressure (TMP). The permeate samples were returned to the feed tank to hold the feed solute concentration. Then, when the steady state was reached (after the first 60 min), feed and permeate samples were taken every 10 min for analysis. Individual ion concentration was measured as it is described in Section 2.6. The permeance of the membrane (P, L m^−2^ h^−1^ bar^−1^) was calculated from the solution flux and the applied pressure [23].
(1)P=(mρ)S·t·p
where m is the sample weight (g), ρ is the density value of the solution at room temperature (g L^−1^), S is the effective surface of the membrane (m^2^), p is the pressure applied during the crossflow filtration (bar), and t is the experimental time (h).

Salt rejection was calculated by measuring the conductivity of the feed ($C_{f}$) and the permeate ($C_{p}$) as is indicated in the following equation [15]:(2)$$\%R=\left(1-\frac{C_{p}}{C_{f}}\right)\cdot 100$$

Both (permeance and salt rejection) were calculated with an average of at least six measurements (relative error ˂ 5%), and by repeating the experiments 3 times with 3 different membrane coupons.

#### 2.4.2. Electrodialysis Experiments

The ED schematic diagram and the stack configuration are represented in Figure 2b. Therein, 5 cation-exchange commercial membranes and 4 anion-exchange membranes (thus, 4 cell pairs) were alternatively arranged between two electrodes. Two different stack configurations were tested, the first one using a commercial cation-exchange membrane (Ralex^®^ CMH-PES, from Mega a.s., Czech Republic) and rAEM, and a second one assembled using only commercial membranes (CMH-PES and AMH-PES). The electrodes were a dimensionally stable electrode (DSE, titanium coated with iridium oxide) for the anode (Inagasa S.A., Barcelona, Spain) and a stainless-steel electrode for the cathode (Tamesanz^®^, Madrid, Spain). The effective area of each membrane was 16 cm^2^. To ensure a tortuous configuration for the solutions path and separate the membranes, allowing complete mixing and air removal in the ED stack [24], the reused polypropylene spacers (0.8 mm thick, 3 mm mesh size) from the end-of-life RO membrane were arranged between the membranes in all the experiments.

In this study, the experiments were carried out at constant voltage in a batch mode at room temperature (~21 °C). The synthetic UWW solution was used as feed for the diluted compartment. Na_2_SO_4_ solution (with a conductivity similar to the diluted solution) was fed in the concentrated chamber. The relation of volume concentrate:dilute (Vc:Vd) was 500:500 mL. A 4 g L^−1^ Na_2_SO_4_ solution was used for both electrode compartments, connected to the same reservoir to avoid pH changes and the potential drop in the ED system. A peristaltic pump HEIDOLPH PD 5206 with a multichannel head circulated the solutions throughout the membranes and anolyte/catholyte chambers, maintaining a uniform flow rate of 20 mL min^−1^. The power supply was an EA-PS 5080-10 A (0–80 V) from EA Elektro-Automatik, Viersen, Germany. The voltage value applied and current variation values were recorded from an Amprobe AM-540-EUR and an Amprobe AM-500-EUR multimeter, respectively. In order to determine the applied operational voltage and the limiting current density (LCD), the same feed solutions were passed through the system. The initial voltage was 0 V and it was increased stepwise up to 30 V, and the resulting current was recorded for each voltage.

In all ED assays, the membranes inside the ED stack were previously equilibrated with the synthetic UWW solution for 24 h. Before the experiments, the working solutions were replaced with new ones and circulated throughout the system for 30 min without an applied electrical current for the homogenization of the solutions. Then, samples were taken from dilute and concentrate reservoirs, analyzed, and recorded as the initial value. Throughout the experiments, samples of concentrate and dilute solutions were taken periodically without exceeding 10% of the total dilute volume variation in each test. Elemental parameters (pH and EC) and concentration of individual ions (Cl^−^, NO_3_^−^, SO_4_^2−^, Na^+^, K^+^, Ca^2+^, and Mg^2+^) were measured for each sample, following the methodology specified in Section 2.6.

The specific energy consumption ($E_{consumption}$, kW h m^−3^) of the process was calculated as [25]:(3)$$E_{consumption}=\frac{U}{V_{d}}\int_{0}^{t}Idt$$
where U is the potential applied to the ED cells (V), I is current (A), $V_{d}$ is the initial volume of the dilute solution (m^3^), and t is the ED operation time (h). In this case, the integration (which is defined as Riemann integral) was solved as the calculation of the area under the curve (plotted *I* vs. *t*) as follows: $I\left(t\right)\approx \sum_{0}^{t}\left[\frac{I\left(t_{f})+I(t_{i}\right)}{2}\left(t_{f}-t_{i}\right)\right]$, where I(t) is a function defined over the positive interval between 0 and the experimental time and $\left(t_{f}-t_{i}\right)$ is the difference in the time of the evaluated range. It should be mentioned that the energy consumption only included the operational energy consumed by the demineralization process evaluated.

The demineralization rate (%DR) indicates the total amount of salt removed, and it was calculated according to the following equation [26]:(4)$$\%DR=\left[1-\frac{\kappa_{t}}{\kappa_{o}}\right]\cdot 100$$
where $\kappa_{o}$ and $\kappa_{t}$ are the initial conductivity and conductivity over time of the dilute chamber, respectively.

To observe and compare the surface morphologies of the rAEM and their stability in the implementation in ED to treat UWW, scanning electron microscopy (SEM) was employed (details in Appendix A).

### 2.5. Irrigation of Lettuce 

Lettuce (*Lactuca sativa* L. var. longifolia) plants, belonging to a Romaine type (Chatelaine), were grown for 7 weeks in a controlled climate growth cabinet (1 m^3^ size). Lettuce seedlings were selected and transplanted when they presented 4 to 5 definitive leaves, between 9 to 15 cm maximum heights. The soil used was standard soil 5M (from Lufa Speyer), and the main characteristics are summarized in Table 1. Standard soils are frequently used for pot experiments in the laboratory and the field. The low level of available N was suitable for the sake of comparison of the treatments. Each pot contained 1750 ± 0.19 g of soil and 374 ± 0.22 g of gravel placed at the bottom as a filter bed. The pots were arranged in a completely randomized design; see Appendix A.

Three different treatments were established with four replicates (plants) each: (i) irrigation with fresh water (tap water, TW), (ii) irrigation with reclaimed water by rNF membrane (IRR), and (iii) irrigation with reclaimed water by ED using rAEM (fertigation, FRT). Throughout the experiment, the temperature (23 ± 1 °C), a photoperiod of 16/8 h (light/darkness), and photosynthetically active radiation (280 µmol m^−2^ s^−1^) were recorded. Each lettuce was irrigated using the drip system installed 0.05 m from the lettuce plant.

After a cultivation period of 49 days, shoots were cut from each plant at the soil level (this can be observed in Appendix A). The growth and yield of the lettuce plants were evaluated through the measurement of head diameter, fresh and dry biomass, and uptake of macroelements.

#### Statistical Analysis

Data were analyzed statistically using a statistical significance level of 0.05 by ANOVA for the main treatments. The comparison between the treatments was evaluated through analysis of means (Tukey–Kramer test; *p* < 0.05) to identify the differences [28].

### 2.6. Analytical Methods

The main parameters for wastewater analysis were measured in the initial synthetic UWW and samples collected from the effluents of NF and ED assays, according to Standard Methods [29], which included EC, pH, anions (Cl^−^, NO_3_^−^, and SO_4_^2−^), cations (Na^+^, K^+^, Ca^2+^, and Mg^2+^), and sodium adsorption ratio (SAR). The conductivity values of solutions were measured by a conductivity meter CM 35 (Crison Instrument, Barcelona, Spain) and the individual ion concentrations by a 930 advanced compact IC Metrohm Ionic Chromatograph.

Prior to the determination of macroelement uptake by the lettuce, the harvested plants were frozen and then freeze-dried using a LyoMicron −80 °C freeze-dryer in order to remove the water present in the lettuce. Hereafter, 100 mg of plant dry mass was taken in a quartz digestion vessel to which 1 mL of boric acid (0.5%), 2 mL of H_2_O_2_ (30%), 0.2 mL of HCl (0.1 M), and 6.8 mL of MilliQ water were added. Subsequently, the samples were digested in the microwave oven (ETHOS One de Milestone) at 190 °C. 

The *SAR* equation is used to predict irrigation water sodium hazards. *SAR* is the ratio of sodium to calcium and magnesium concentration (meq L^−1^) and is calculated as [30]:(5)$$SAR=\frac{\left[Na^{+}\right]}{\sqrt{0.5\;\left(\left[{Ca}^{2+}\right]+\left[{Mg}^{2+}\right]\right)}}$$

However, the *SAR* parameter is not of significant value itself to predict the impact of irrigation water on soil. Either EC or *SAR* affects water infiltration in soil; hence, both must be considered in evaluating water quality for irrigation (Appendix A). In general, sodium hazard increases as *SAR* increases and EC decreases [30].

## 3. Results

### 3.1. UWW Treatment by rNF Membranes

To study the rNF membrane performance in tertiary treatment of UWW (composition detailed in Table 2), the quality of the treated effluent in terms of salinity was evaluated. Figure 3 shows the rNF membrane permeance over the experiment and the total salt rejection (in terms of solution conductivity).

Regarding permeance, we could see two different trends in the curve (Figure 3). The linear trend with a surge over the first 60 min (from 10.8 to 7.8 L m^−2^ h^−1^ bar^−1^) was due to the compaction of the membrane after its transformation. Then, the region where the steady state was achieved (after 60 min) reached the permeance value of 4.89 L m^−2^ h^−1^ bar^−1^ (with an error <1%) after 120 min. In addition, as it is represented in Figure 3, the total salt rejection was maintained at around 80% (decrease in electrical conductivity) during the experiment. This result is comparable with a commercial NF membrane performance [4,30].

At 140 min of crossflow filtration, samples were taken and analyzed. The rNF membrane showed a high selective rejection of SO_4_^2−^ ions (>96%) and very high calcium and magnesium separation coefficients (>93%), in concordance with preliminary characterization tests presented by García-Pacheco et al. [15]. Regarding the rejection of monovalent ions, it showed more significant monovalent separation coefficients: 80.02% for Cl^−^, 66.44% for NO_3_^−^, and 74.54% for Na^+^.

### 3.2. UWW Treatment by ED Applying rAEM 

#### 3.2.1. Determination of the Operating Voltage and LCD

LCD is a critical operating parameter that controls, among other parameters, the optimal demineralization efficiency [31]. The current–voltage curve was experimentally measured in the stack configuration assembled with CMH-PES and the prepared rAEM to determine the LCD, using the UWW as feed solution (Figure 4a). Typically, the current increases linearly at low voltage (Ohmic’ s region), then the increase rate reduces to reach a plateau (LCD region), and finally, the current density increases again (over-LCD region). However, measurements in a multi-cell stack often do not show a clear indication of the slope-changing point. Therefore, the Cowan–Brown method was also applied to verify the LCD value by plotting the overall resistance versus the reciprocal current density (Figure 4b) [32].

Based on the intersection points in Figure 4a, the LCD was identified as equal to 1.90 mA cm^−1^ at the voltage of 8.77 V (2.19 V/cell pair). The same LCD value was defined by the resistance −1/I graph, in which the LCD was considered as the lowest point on the curve as shown in Figure 4b (show up with the arrow). The boundary layer resistance drastically increases in the LCD region because of the complete depletion of the salt at the membrane surface facing the dilute solution [33]. Above LCD, a non-desired phenomenon of water-splitting occurs, which affects drastically the efficiency of the process and can provoke irreversible damage to ion-exchange membranes due to pH changes [34].

Therefore, in this study, the experiments were carried out below the LCD to maximize current efficiency, minimize the boundary layer effect caused by the concentration gradient, and compare testing results. The literature considers the range between 60 and 80% of the limiting voltage the safer operating voltage [34,35]. Thus, the experiments were run at around 80% of the limiting voltage (1.7 V/cell pair).

#### 3.2.2. rAEM Evaluation

To study the performance of rAEM in UWW remediation, the stack was assembled by 5 CMH-PES and 4 rAEM (thus, 4 cell pairs) at a working voltage of 1.7 V/cell pair. The DR percentages were calculated based on the measured dilute conductivity. The relationship between DR and energy consumption is shown in Figure 5. Additionally, the performance of the rAEM in the ED system to demineralize the synthetic UWW was compared with the AMH-PES testing under the same conditions (i.e., the stack assembled with 4 cell pairs, a working voltage of 1.7 V/cell pair, and Vc:Vd of 500:500 mL).

As can be seen in Figure 5, a linear tendency of consecutive DR rise by increasing the energy consumption of the system was observed. In the case of the rAEM, an increase in the energy consumption needed for the demineralization process above 60% of the DR was noticed. The evolution of the demineralization process is correlated with the system resistance (SR). The SR arises from the intrinsic resistance of the membranes inside the ED stack and the resistance of the treated solutions [36]. Thus, the latter hindering ion migration can be attributed to the ion depletion attained for the dilute solution. In this case, the good balance of the macronutrients (corresponding with the water quality for irrigation regulated by WHO) was achieved at 60% of the DR for both systems (stack assembled by rAEM and the stack assembled by AMH-PES), which is further discussed in the corresponding section (Section 3.3). Therefore, 60% of DR was set as the completion of the ED experiments.

Additionally, the energy consumption of the stack assembled with rAEMs was less than 1.5 kW h m^−3^ to reach 60% of DR. Under the same conditions, the stack assembled using only commercial membranes consumed 1.06 kW h m^−3^ to reach the same desalination rate. The slightly higher energy consumption of the rAEMs can be attributed to their higher electrical resistance in comparison with the commercial Ralex membranes. It can be considered that the rAEMs operated with a good level of permselectivity, considering the small difference in the energy consumption between both systems. Furthermore, the slightly higher energy consumption of recycled membranes could be balanced by a lower economic cost associated with the production of such membranes [37]. Surface SEM micrographs were employed to observe the rAEM stability after the experiment. No significant differences could be noticed from the images in Appendix A. Further, pinhole and crack formation were not detected by the SEM analysis.

Overall, membrane recycling is a more sustainable approach than landfill disposal of end-of-life membranes, and it has been demonstrated that the production of recycled membranes results in a lower water and carbon footprint than the production of new membranes [38,39].

### 3.3. Water Quality for Crop Irrigation

Conductivity is a very important water quality factor for crop production as a high conductivity causes the inability of plants to compete with ions in the soil solution and water [40]. In addition to conductivity, sodium imbalance in irrigation water can have a substantial impact on crop production. When irrigation water has high sodium content relative to the calcium and magnesium contents (i.e., a high SAR value), water infiltration decreases [41]. Thus, the main parameter compared in this study was the SAR value along with the EC of the treated effluents. 

The initial conductivity of UWW was higher than the limits for use in agriculture without severe restrictions (>3 dS m^−1^, [4]). After the rNF treatment, as expected, the conductivity dropped from 4.90 to 0.98 dS m^−1^ (Table 2) since the NF membranes usually provide good retention of inorganic salts, especially if multivalent ions are involved [10]. According to WHO guidelines (Appendix A), which are connected to the crop sensibility and the content of salts, the rNF effluent falls in the category of “Slight to moderate” concerning the water infiltration, as the calculated SAR value was 6.85 and EC < 1 dS m^−1^.

On the other hand, the ED process using 16 cm^2^ of rAEM area (64 cm^2^ of rAEM in total) to treat 0.5 L of the UWW was carried out in a mean time of 8 h. The effluent quality obtained at that time, corresponding with a 60% value of DR, could be considered in the category of “Slight to moderate” concerning the water infiltration because the SAR value was 6.01 (in the 6–12 range) and EC was <2 dS m^−1^ (500–1900 µS cm^−1^). Eventually, the ED treatment showed a reasonable extraction for all ions in the solution without compromising the balance ratios and ensuring adequate-quality water for reuse (Table 2).

As a result, both tested recycled membranes showed adequate potential in wastewater treatment for crop irrigation purposes in terms of conductivity and SAR value.

### 3.4. Lettuce Yield and Macronutrient Uptake in Dry Weight

Lettuce is a vegetable crop that has high nutritional and health value and is best consumed fresh. Fresh lettuce leaves contain about 91–96% water [42]. Figure 6 shows the average values of the analyzed lettuce fresh wight samples (a) and water content for each treatment, (b) using the different water qualities obtained.

No statistically significant differences were found in the biomass-fresh weight according to the treatments TW, IRR, and FRT, which ranged from 34 to 37 g. However, the water content (WC) percent was altered between TW and FRT treatments, showing a statistically significant difference (*p* < 0.05). Although the WC was not drastically affected by the different nutrient solution concentrations (IRR and FRT), the WC value for lettuces irrigated by IRR water was below the normal range for this crop (i.e., 91–96%). This effect might be attributed to the influence of the nutrient balance of the different effluents, applied as treatments, on water uptake by plants [42].

Individual macronutrients such as total nitrogen, potassium, calcium, and magnesium of leaves of the lettuce were chemically analyzed (Table 3). In addition to the different nutrient availability of each treatment studied, no significant differences in nutrient absorption were observed between them.

Vegetables, especially leafy ones, represent the major sources of dietary nitrate intake, owing to their nitrate accumulation capacity. Nitrate and nitrite (which can be formed as an intermediate product of nitrate reduction) are toxic to human health. Therefore, it is essential to keep a low nitrate concentration in the edible parts of crop plants [43]. For the reused wastewater treatments, proper values of nitrogen uptake by lettuce leaves (shown in Table 3) were achieved in compliance with the literature data (range from 1.13 to 5.02% N in dry weight), which was summarized by M. Petek et al. [42]. Potassium content in lettuce dry weight in the three treatments ranged from 4.09 to 4.91%. The highest potassium content was determined in FRT treatment, which is almost twice higher than that the reported by M. Petek et al. [42] but in agreement with M. R. Broadley et al. [44] who reported 4.5% K DW in lettuce. Calcium content in lettuce dry weight in treatments ranged from 1.48 to 1.71% Ca. The highest calcium content was found in TW treatment, which is in concordance with the results reported by M. Petek et al. [42] (i.e., the highest calcium content (1.42% Ca) in the treatment with no fertilization applied). Behavior could be explained by the different concentrations of ammonium-N. High doses of ammonium-N cause impairment of other nutrient absorption (e.g., Ca) due to competition between NH_4_ and Ca^2+^ cations [45]. The average magnesium content in lettuce dry matter was 0.27% Mg, which is lower than the 0.52% Mg DW value obtained in Ref. [42] but falls within the range of 0.15 to 0.35% Mg DW reported in the literature [42].

## 4. Conclusions

Different membrane processes can be integrated into wastewater treatment systems, considering the challenging properties of saline wastewater. To make membrane processes an economically attractive and viable alternative, especially in the context of sustainability, two different recycled membranes in treating saline wastewater from an urban WWTP were validated in this work.

The rNF membrane showed a high selective rejection of divalent ions (i.e., SO4^2−^ (>96%); Ca^2+^ and Mg^2+^ (>93%)).Comparison between rAEM and commercial anion-exchange membranes (Ralex^®^) showed a suitable demineralization rate for irrigation of crops without compromising the power consumption.Both tested recycled membranes showed adequate potential in wastewater treatment for crop irrigation purposes in terms of conductivity and SAR value. No significant differences in individual macronutrients such as total N, P, Ca, and Mg of leaves of the lettuce of each treatment studied were observed.

Overall, this research showed, for the first time, the successful performance of recycled membranes for treating saline urban wastewater and its application for crop irrigation. In addition, this study showed that membrane recycling is a technically viable process that increases the sustainability of water separation processes and enables the valorization of wastewater for irrigation purposes, which conveys the strategy of the European Commission regarding the transition to a circular economy.

## Figures and Tables

**Figure 1 membranes-12-00746-f001:**
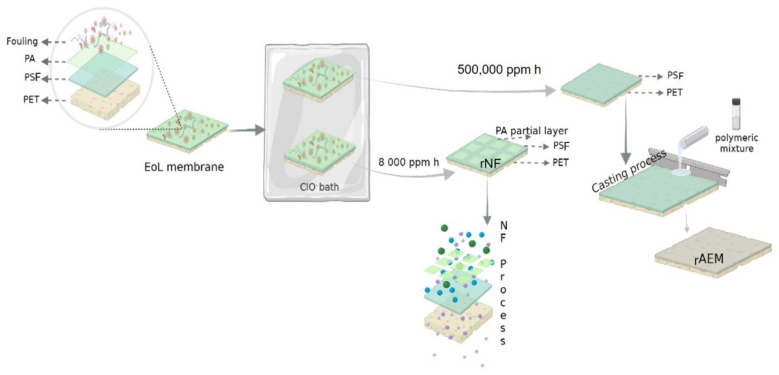
Schematic representation of membrane preparations (rNF and rAEM membranes). PA, polyamide; PSF, polysulfone; PET, polyester.

**Figure 2 membranes-12-00746-f002:**
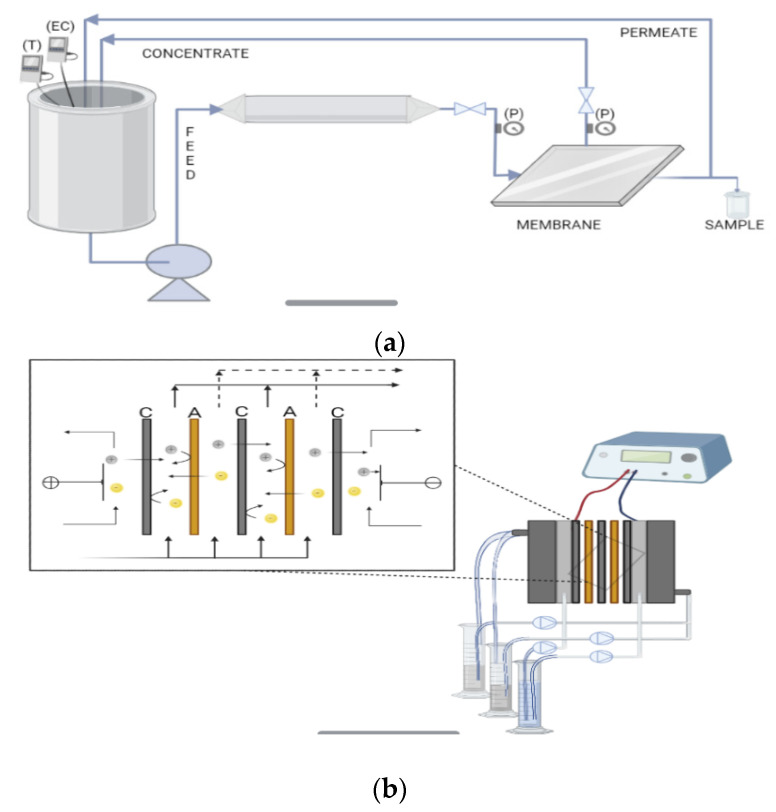
Diagram of (**a**) NF system (T, temperature; EC, electric conductivity; P, pressure) and (**b**) ED system (C, cation-exchange membranes; A, anion-exchange membranes).

**Figure 3 membranes-12-00746-f003:**
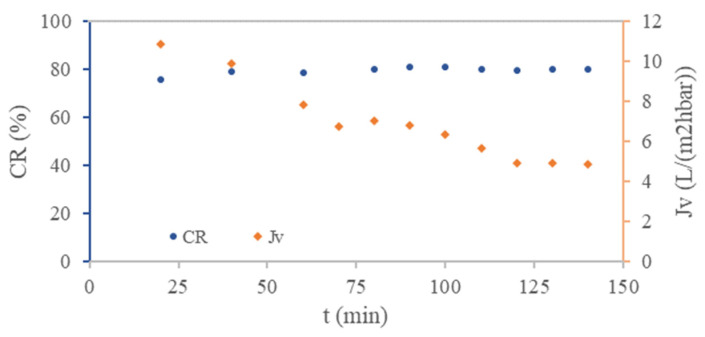
Conductivity rejection (CR) and permeance (Jv) in the function of time (t) for the recycled NF membrane.

**Figure 4 membranes-12-00746-f004:**
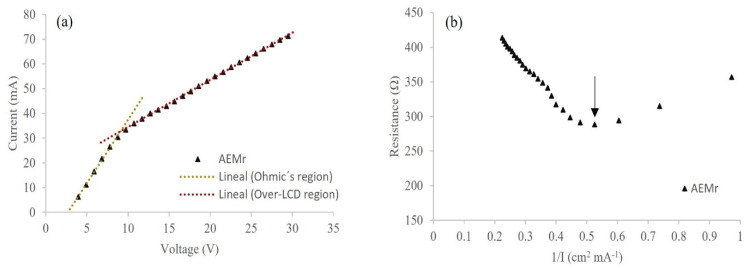
(**a**) Current-voltage and (**b**) Cowan-Brown method for the determination of LCD with recycled anion-exchange membranes (rAEMs) (flow rate of 20 mL min^−1^ and 4 cell pairs).

**Figure 5 membranes-12-00746-f005:**
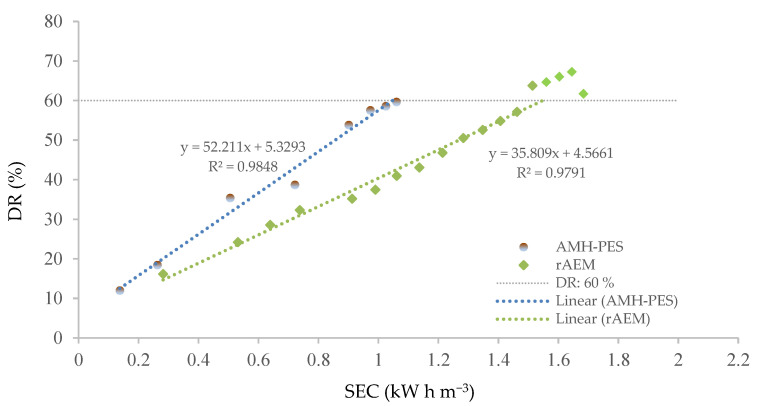
Comparison of demineralization rate (DR, %) of AMH-PES and rAEM based on specific energy consumption (SEC, kW h m^−3^) by electrodialysis treatment.

**Figure 6 membranes-12-00746-f006:**
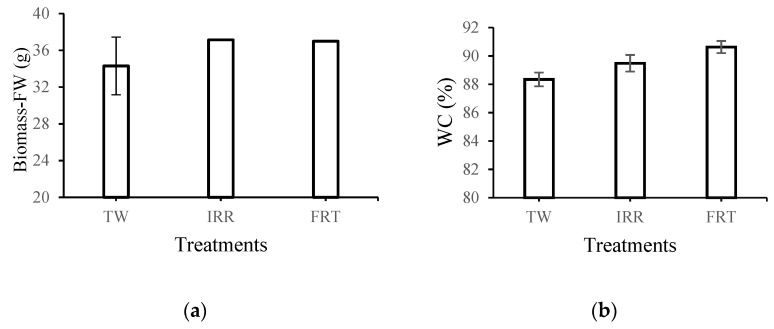
Comparison of the different treatments (TW: tap water; IRR: treated wastewater by NF; FRT: treated wastewater by ED) (**a**) according to the average biomass fresh weight (FW) and (**b**) water content (WC, %) in lettuce leaves. Error bars are reported as standard errors (SEs).

**Table 1 membranes-12-00746-t001:** Chemical and physical characteristics of standard soil 5 M ^1^.

Parameters		Standard Soil 5M
Soil type		Sandy loam
Sampling date		20 July 2021
pH value (0.01 M CaCl_2_)		7.4 ± 0.1
Organic carbon (% C)		0.88 ± 0.18
Nitrogen (% N)		0.11 ± 0.03
Max. water-holding capacity (g/100 g)		41.8 ± 5.3
Weight per volume (g/1000 mL)		1219 ± 88
Cation-exchange capacity (meq/100 g)		8.5 ± 0.25
** *Particle size distribution (mm) according to USDA (%)* **		
**<0.002**	**0.002–0.05**	**0.05–2.0**
11.9 ± 1	31.6 ± 3.2	56.5 ± 3.3

^1^ Data taken from the supplier (Lufa Speyer) [27].

**Table 2 membranes-12-00746-t002:** Characteristics of the synthetic solution and the effluents of NF and ED technologies.

Water Source	Conductivity (dS m^−1^)	Ionic Compound (ppm)					
		Cl^−^	NO_3_^−^	Na^+^	K^+^	Ca^2+^	Mg^2+^
Synthetic UWW	4.90 (±0.1)	1224.8	68.21	694	47.20	209	102
Permeate of rNF	0.98 (±0.02)	249	24.20	153	14.50	14.10	6.86
Product of ED (*rAEM*–*CMH*-*PES*)	1.93 (±0.1)	390	22.90	320	23.60	52.90	29.20

**Table 3 membranes-12-00746-t003:** Macronutrient concentrations of leave-in dry weight.

	NTK	IN	K	Ca	Mg
	(%)	(%)	(%)	(%)	(%)
**TW**	1.03	0.11	4.10	1.71	0.25
**IRR**	1.20	0.12	4.42	1.48	0.28
**FRT**	1.24	0.11	4.91	1.68	0.29
CV (%)	0.22	0.22	1.16	0.48	0.10

CV%: coefficient of variation; NTK: total nitrogen Kjeldhal; IN: inorganic nitrogen (NO_3_^−^ + NO_2_^−^); TW: tap water; IRR: treated wastewater by NF; FRT: treated wastewater by ED.

## Data Availability

Not applicable.

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
