# Peer review of "Validation of Recycled Nanofiltration and Anion-Exchange Membranes for the Treatment of Urban Wastewater for Crop Irrigation"

_membranes, 2022, doi:10.3390/membranes12080746_

Round 1

Reviewer 1 Report

The review comments for membranes-1835077

 Wastewater treatment is essential to produce potable water and the prevention of human interference with the natural ecosystem. The membrane process has been considered as the most efficient method for treating wastewater while reusing the available resources. In this work, the authors propose recycling polluted or defective RO membranes through the manufacturing of regenerated nanofiltration and anion exchange membranes. The ultrafiltration membrane obtained from end-of-life RO membrane was employed as the membrane matrix and then surface-modified. As demonstrated by the results, the membranes developed have a substantial desalination efficiency. Interestingly, freshwater products were employed for planting lettuce crop.. Overall, the paper is well-organized and would be suitable for publication following minimal revisions. Listed below are a few minor questions for the author's consideration.

1.      1. It is unclear why anion exchange membrane rather than cation exchange membrane was produced for electrodialysis desalination.

2.      Page 4, line 178-179, it is proposed “This membrane (Ralex® AMH-PES) was selected as a reference as long as it has a heterogeneous structure…” Typically, heterogeneous ion exchange membranes are produced by combining ion exchange resin powder with a polymer matrix. The membrane developed in this study is not, however, in this structure, which should be carefully examined.

3.      It is important to provide specific instructions on how to mark the completion of ED experiments.

4.      The results of the lettuce crop could be supplemented furtherly for better understanding, such as the photography of the crop during the evaluation.

Reviewer 2 Report

The main objective of this paper is to investigate the potential of recycled nanofiltration and anion-exchange membranes for the treatment of high salinity urban wastewater. The paper can be considered for acceptance after revision based on these comments. The methodology for calculating specific energy consumption is very little;  a much more lengthy on flow of each equation is needed. Moreover, did the author consider the energy for reusing membrane in energy calculation?

Reviewer 3 Report

The manuscript written by Pernia et. al prepared NF membranes and AEMs from the recycled RO membranes taken from the spiral wound membrane modules based on 2 previous papers of their group and used these 2 kinds of membranes to treat real urban wastewater samples. The results are very interesting, that the recycled membranes can be reused as NF or AEM membranes and remove all ions, particularly, Na+ ions to reduce the conductivity. The treated water can be used for irrigation purposes to cultivate lettuce. I find this work is technically sound, the experimental design is careful and the manuscript is well written. This paper can be published in Membranes after addressing the following points:

1. The authors mentioned high salinity in the title of the paper, claiming that their membranes are validated to treat these “high salinity” waters, and defined it later in the introduction, saying a salt concentration of 1%-3.5% is high-salinity. However, the urban wastewater (UWW) samples they tested only have 0.2 wt% of total salt content. This is far from being considered as “High-salinity”. As many researchers have pointed out, when the feed NaCl concentrations are higher (at 1%- 3.5%), the rejections of the NF membranes go down quickly due to charge screening. (https://doi.org/10.1038/s41893-020-00674-3) (https://doi.org/10.1016/0011-9164(96)00024-0). The effectiveness of their membranes under the conditions they claimed remained obscure.

Please either (1) revise the title by removing the “High-salinity” claim, mention in the introduction that this work only investigated the low salt concentration UWW as a prove-of-concept, and mention the effect of feed ionic strength on NF rejections (AEM removal rate) by adding proper references, and mention the upper-limit of the NaCl level that these 2 membranes can handle; or (2) Add additional experiments of NF and AEM to show that these membranes actually works in the specified salinity ranges (1 wt% -3.5 wt%).

2. The definition of “permeability” is not correct, it should be “permeance”. The unit of permeance is L m-1 h-1 bar-1. The unit of permeance is L m-2 h-1 bar-1. It is an international scientific research community consensus, that permeability is an intrinsic property of the membrane material (which equals permeance times membrane thickness), and permeance is flux divided by driving force. Please correct. Please refer to this JMS paper: Journal of Membrane Science 348 (2010) 346–352.

3. The authors are claiming that their rNF membrane performance is comparable to the NF-90, but there are no control experiments that tested the two membranes under the same conditions. The reviewer suggests that the authors can add a table, listing out the performances of the NF-90 membrane and your membranes under the same testing conditions.

4. Please provide the rejection for Na+ and Cl- and their concentrations in Table 3. The authors only provided the SAR values but the rejection of NaCl is still unclear. Given that the authors claimed the sodium ions are the ions that needed to be removed, it is better to provide this information.  
